# Ertapenem Supplemented Selective Media as a New Strategy to Distinguish β-Lactam-Resistant Enterobacterales: Application to Clinical and Wastewater Samples

**DOI:** 10.3390/antibiotics12020392

**Published:** 2023-02-15

**Authors:** Alexandre Bourles, Malia Kainiu, Damaris Ukeiwe, Nina Brunet, Camille Despaux, Antoine Biron, Ann-Claire Gourinat, Cyrille Goarant, Julien Colot

**Affiliations:** 1Institut Pasteur de Nouvelle-Calédonie, Pôle de Bactériologie, Groupe de Bactériologie Médicale et Environnementale, CEDEX BP 61, Noumea 98845, New Caledonia; 2CRESICA Department, Université de la Nouvelle-Calédonie, CEDEX BP R4, Noumea 98851, New Caledonia; 3Centre Hospitalier Territorial Gaston-Bourret, Laboratoire de Biologie Médicale, CEDEX BP J5, Noumea 98849, New Caledonia

**Keywords:** antimicrobial resistance, carbapenemase-producing-carbapenem-resistant-enterobacterales, ertapenem, wastewater, culture-based approach, New Caledonia

## Abstract

The increase in carbapenem-resistant Enterobacterales (CRE) is mostly driven by the spread of carbapenemase-producing (CP) strains. In New Caledonia, the majority of carbapenemases found are IMP-type carbapenemases that are difficult to detect on routine selective media. In this study, a culture-based method with ertapenem selection is proposed to distinguish non-CRE, non-CP-CRE, and CP-CRE from samples with very high bacterial loads. Firstly, assays were carried out with phenotypically well-characterized β-lactam-resistant Enterobacterales isolates. Then, this approach was applied to clinical and environmental samples. Presumptive CP-CRE isolates were finally identified, and the presence of a carbapenemase was assessed. In a collection of 27 phenotypically well-characterized β-lactam-resistant Enterobacterales, an ertapenem concentration of 0.5 µg·mL^−1^ allowed distinguishing CRE from non-CRE. A concentration of 4 µg·mL^−1^ allowed distinguishing CP-CRE from non-CP-CRE after nine hours of incubation. These methods allowed isolating 18 CP-CRE from hospital effluents, including the first detection of a KPC in New Caledonia. All these elements show that this cost-effective strategy to distinguish β-lactam-resistant Enterobacterales provides fast and reliable results. This could be applied in the Pacific islands or other resource-limited settings, where limited data are available.

## 1. Introduction

The worldwide increase in antimicrobial resistance (AMR) in clinically relevant microorganisms is becoming critical. The global burden associated with drug-resistant infections assessed across 88 pathogen–drug combinations in 2019 was an estimated 4.95 million deaths, of which 1.27 million were directly attributable to drug resistance [1]. Resistance to fluoroquinolones and β-lactams (i.e., penicillins, cephalosporins and carbapenems), often first-line drugs for empirical therapy of severe infections [2], accounted for more than 70% of deaths attributable to AMR across pathogens [1].

Enterobacterales belonging to *Escherichia*, *Klebsiella*, *Enterobacter*, *Serratia*, *Citrobacter* genera are common pathogens causing a variety of severe infections, including bloodstream infections, pneumonia, complicated urinary tract infections, and complicated intra-abdominal infections. Carbapenem-resistant Enterobacterales (CRE) are among the top tier of the WHO list of antibiotic-resistant “priority pathogens” that pose the greatest threat to humans [3]. According to the United States of America Center for Diseases Control and Prevention definition, Enterobacterales that test resistant to at least one of the carbapenem antibiotics or produce a carbapenemase (an enzyme that cleaves carbapenems and can make the bacteria resistant to carbapenem antibiotics) are called CRE. The resistance of CRE to carbapenems is generally based on either or both of two mechanisms: carbapenemase production (carbapenemase-producing-carbapenem-resistant-Enterobacterales; CP-CRE) or the combination of structural mutations with the production of other β-lactamases (non-CP-CRE), such as AmpC β-lactamase (AmpC) associated with porin change and/or efflux pump and extended-spectrum β-lactamase (ESBL), which also exhibit low-level carbapenem-hydrolyzing activity and can confer some degree of carbapenem resistance [4].

AMR is also a critical global health threat, and Pacific Island Countries and Territories (PICTs) are not spared. Although available data on AMR in this region are limited to a few major centres [5], AMR constitutes a real public health problem in these islands [6,7]. New Caledonia is a French archipelago located in the tropical southwest Pacific Ocean, about 1200 km East of Australia. It is composed of a main island, where most of the 271,407 inhabitants counted in 2019 live, and smaller islands. The territory has frequent air links with Australia, the site of numerous medical evacuations, and mainland France, with which population transfers are very frequent. In this archipelago, IMP-type carbapenemase-producing isolates were the first CP-CRE identified in 2013. Since then, the number of IMP-type carbapenemase (encoded by *bla_IMP-4_* gene) carriers increased from 2013 (4 isolates) to 2018 (33 isolates found) [8]. IMP-4 carbapenemases become the predominant CP-CRE type in Australia while continuing to spread in the Asia Pacific region [9,10]. Interestingly, the vast majority of CP-CRE found in New Caledonia are also IMP-4 [8].

Among several methods for identifying CRE, the culture-based method with chromogenic media is most used for the initial detection of CRE. The chromogenic media is a sensitive, convenient, and relatively low-cost way of identifying CRE [11]. However, some carbapenemases such as IMPs exhibit a low minimum inhibitory concentration (MIC) to carbapenems, making it difficult to detect these using commercial carbapenem-supplemented agar plates [12,13].

In New Caledonia, ChromID ESBL (bioMérieux, Marcy-l’Étoile, France), a chromogenic medium designed for the isolation of ESBLs which includes cefpodoxime and chromogenic dyes is used to detect CRE. On this medium, pink and green colonies belong to *Escherichia coli* and KESC (*Klebsiella*, *Enterobacter*, *Serratia*, and *Citrobacter*) groups, respectively. However, this medium prevents any distinction between non-CRE, non-CP-CRE, and CP-CRE [14]. Given the high diversity and abundance that can be found on these plates when a high bacterial load is inoculated, this study was designed to establish a culture-based approach for detection and differentiation of these three resistance mechanisms. In accordance with the European Committee on Antimicrobial Susceptibility Testing (EUCAST), ertapenem, the carbapenem drug with the highest sensitivity for CRE detection [15], is used. Based on susceptibility to ertapenem, the objective of this study was to develop a culture-based approach using ertapenem-supplemented selective media to distinguish presumptive non-CRE, non-CP-CRE, and CP-CRE in a high bacterial load sample. Firstly, assays were carried out with isolated phenotypically well-characterized Enterobacterales resistant to β-lactams to determine growth patterns in the presence of ertapenem of different resistance mechanisms (ESBL, AmpC, Carbapenemases). Then, this culture-based approach is applied to environmental and clinical samples. Results on clinical samples were compared to data of the medical bacteriology laboratory of the Centre Hospitalier Territorial (CHT), the main and reference hospital of the archipelago. Presumptive CP-CRE isolates were finally identified using MALDI-ToF MS, and the presence of a carbapenemase was assessed with a synergy test (EUCAST, 2022) and confirmed with a commercial immunochromatography test (ICT). Finally, based on first results and to improve turnaround time for diagnosis, detection of carbapenemases (using ICT) after only nine hours of incubation in ertapenem-supplemented selective media was performed.

## 2. Results

### 2.1. Implementation of the Technique

Growth curves for the 27 controls strains (Table 1) are summarized in Figure 1. At 24 h, in each well where a growth was measured, culture was pure and viable. All isolates displaying growth at a 4 µg·mL^−1^ concentration were confirmed as CP-CRE using ICTs. For non-CP-CRE, growth was observed in medium without ertapenem and supplemented with 0.5 µg·mL^−1^ of ertapenem, but not with 4 µg·mL^−1^. However, ertapenem had a significant bacteriostatic effect at 0.5 µg·mL^−1^ at each incubation time point. Details between AmpC-producing Enterobacterales and ESBL + AmpC-producing Enterobacterales are presented in Appendix A. For non-CRE, growth did not occur at any ertapenem concentration. The results of CP-CRE according to species and class of carbapenemase are shown in Appendix A.

### 2.2. Application to Clinical Samples

Our method was then tested on 171 isolated colonies from 30 ChromID ESBL agar plates and results are presented in Table 2. No CP-CRE was isolated from any of the samples, and a high proportion of presumptive non-CRE was found (*n* = 122; 71%), with 48 isolates identified as presumptive non-CP-CRE (29%). Comparison with the CHT laboratory results showed a high concordance of results (83%). For only three samples, different phenotypes were detected by our method in comparison to the CHT laboratory results. For these three non-concordant samples, the occurrence of an AmpC β-lactamase was detected by the CHT laboratory, whereas growth over time suggested presence of non-CRE. Additionally, for one sample, an additional phenotype (Non-CRE) was detected in comparison to the CHT laboratory results.

### 2.3. Application to Environmental Samples

The method was also tested on 104 isolated colonies from wastewater samples MED3, MED4, MED5 and MED6. The results for all presumptive resistance phenotypes are presented in Table 3. In all samples, a high proportion of presumptive non-CRE was found. This rate was higher than 80% for the MED4, MED5, and MED6 samples. Considering presumptive CRE, twenty-four isolates were able to grow at both concentrations of ertapenem, and growth curves for these presumptive CP-CRE are shown in Figure 2 (representative growth plots of non-CRE and non-CP-CRE are available in Appendix A). For seven isolates (MED3-1, -2, -3, -4; MED5-31, MED6-25, -26), no bacteriostatic effect of ertapenem was observed compared to the control. For fourteen isolates (MED5-29, -30), a bacteriostatic effect was observed in LB supplemented with 4 µg·mL^−1^ of ertapenem compared to the control. A significant growth in this medium could be visualised from 6 (MED5-30) to 9 (MED5-29) hours of incubation onwards. Finally, growth occurred after 24 h at each concentration; however, a bacteriostatic was sometimes still observed in LB supplemented with 4 µg·mL^−1^ of ertapenem. For MED3-5, MED4-6, and MED5-31, a slight bacteriostatic effect was observed in LB supplemented with 4 µg·mL^−1^ of ertapenem in comparison to the control. The 12 presumptive CP-CRE were identified using MALDI-ToF MS: five isolates belonged to *Citrobacter*, one to *Enterobacter*, four to *Klebsiella* and two to *Raoultella* genera (Table 4). All 12 isolates were multi-drug resistant bacteria (Table 4). For 10 of these, the synergy tests confirmed the production of a carbapenemase and the presumptive occurrence of an IMP (CLT contact) confirmed by the ICT. For two *K. pneumoniae* isolates, the synergy test could have suggested an ESBL (clavulanic acid recovered and inhibition diameter of ertapenem was borderline of the cut-off), but the ICT confirmed the presence of a KPC-carbapenemase (Table 4; Appendix A).

### 2.4. Growth over Time and Carbapenemase Class Confirmation after 9 Hours of Incubation

Based on our results, carbapenemase detection of isolates (*n*= 64) from MED7 were carried after a shorter 9 h incubation. Growth over time in the presence of ertapenem of presumptive CP-CRE in MED7 is presented in Figure 3, and all results are summarized in Table 5. The results for all presumptive resistance phenotypes (detected after 24 h of incubation) are presented in Table 3. Fifteen isolates belonging to *Klebsiella pneumoniae* species were considered as presumptive CP-CRE after nine hours of incubation (71% of presumptive CP-CRE) and among these isolates, occurrence of an IMP-carbapenemase was confirmed for ten isolates of thirteen tested (77%). Six isolates (four *Klebsiella* spp. and two *Serratia* spp.) were considered as presumptive CP-CRE after 24 h of incubation. For these isolates, a carbapenemase was confirmed using a synergy test and an ICT: these isolates harboured an IMP for one *Klebsiella oxytoca;* in addition, a KPC was detected (Table 4).

## 3. Discussion

### 3.1. Supplemented Ertapenem Selective Media

The development and spread of antibiotic-resistant bacteria is a global problem recognized by health and scientific organizations as one of the major public health challenges of the 21st Century. Among them, CP-CRE are considered “priority pathogens” [2] and for their detection, commercial selective media are classically used by diagnostic laboratories [11,12,13]. However, some CP-CRE that exhibit a low MIC to carbapenems may remain undetected by this approach [12]. This is the case in New Caledonia, where, since 2013, IMP carbapenemases have been found in a vast majority of cases [8]. Distinguishing non-CP-CRE and CP-CRE has important infection-control implications because genes encoding carbapenemases are generally located on multi-drug mobile genetic elements (i.e., plasmids, transposons, and insertion sequences) and are easily transmissible to other Gram-negative organisms [16]. Furthermore, some reports suggested that CP-CRE might be more virulent than non-CP-CRE [17]. Additionally, with novel antimicrobials agents, the distinction between CP-CRE and non-CP-CRE will be highly important for antimicrobial stewardship programs. In this sense, our study proposed rapid, accurate, cost-effective, and standardized culture-based method with ertapenem selection to distinguish non-CRE, non-CP-CRE, and CP-CRE from a sample where high diversity and abundance are found. Carbapenemases are enzymes that degrade the antibiotic, which makes it necessary to work on each isolated colony individually to avoid ectopic complementation of sensitive strains from excreted enzymes. As used in others studies [18,19], the European breakpoint value of MIC for ertapenem (0.5 µg·mL^−1^) allowed the distinction between CRE and non-CRE isolates. In our study, among CRE, isolates which were able to growth at 0.5 µg·mL^−1^ of ertapenem but not at 4 µg·mL^−1^ of ertapenem were classified as non-CP-CRE. CP-CRE were isolates which have succeeded in growing until 4 µg·mL^−1^ of ertapenem. A perfect match was found between the resistance mechanism identified at the phenotypic level and the affiliation to presumptive carbapenem resistance mechanism.

In 2016, Tamma et al. [20] used MIC cutoffs to distinguish CP-CRE from non-CP-CRE. They showed that if a CRE isolate remained susceptible or intermediate to some carbapenems (which is the case for IMP-type carbapenemases), an ertapenem MIC of 0.5 µg·mL^−1^ or greater represented the carbapenem MIC’s that was best to discriminate between CP and non-CP-CRE. Our results do not follow this pattern; indeed, all non-CP-CRE and CP-CRE were able to growth at 0.5 µg·mL^−1^ of ertapenem. Such differences could also be explained by the collection of isolates used (generally in relation with local epidemiology, which is also our case). Their study [20] suggested that it would prove useful to repeat these analyses with more isolates to increase the accuracy of carbapenem MICs for distinguishing CP-CRE from non-CP-CRE. Such differences could also be explained by the techniques used. Indeed, in their study, Tamma et al. [20] used two reference methods to determine MIC: reference broth microdilution and E-test. Our method was not designed to determine MICs; moreover, when results of growth over time in the presence of ertapenem are compared with MICs obtained with VITEK^®^ 2 system (Table 1), these differences are found again. In our study, confirmed CP-CRE with ertapenem MICs identified as below 0.5 or equal to 2 µg·mL^−1^ of ertapenem using VITEK^®^ 2 system still grew successfully in our microplate test in wells at 4 µg·mL^−1^ of ertapenem after 24 h of incubation, suggesting that no comparison was possible between ertapenem phenotypes obtained from our growth curves and the VITEK^®^ 2 system.

From a practical point, this method has several advantages: a large number of isolated colonies can be screened rapidly, so this is a cost-effective method and an easily implemented approach in a microbiology laboratory. In 2020, Fergusson et al. [21] highlighted that Pacific island nations will need to prepare for CP-CRE emergence, including the ability to reliably detect CP-CRE in laboratories. This method responds to this request, and could be applied in other PICTs and other low-resource settings. In these countries, for life-threatening infections, broad-spectrum therapies, such as the carbapenems, will likely be used more frequently because of the lack of robust surveillance data or limited laboratory capacities. For example, in 2020, Foxlee et al. [22] described pathogen occurrence and antibiotic resistance in specimens cultured at Vila Central Hospital (VCH), the main referral hospital in Port Vila, the capital of Vanuatu. Sixty-one percent of isolates (373/607) belonged to *Escherichia*, *Citrobacter*, *Klebsiella*, *Enterobacter*, *Serratia* genera, for which there were no data on carbapenem resistance. In this case our technique could be applied in order to generate data on carbapenem resistance in these medical settings.

The use of a liquid culture saves time in terms of results compared to agar plating. For diagnostic application, detection of presumptive CP-CRE using an ICT was performed after a 9 h incubation. Our results indicated that detection was possible, but the sensitivity decreased to around 70% among the isolates tested. All carbapenemases were identified at T24. This step is therefore necessary, and ICT appears also essential in order to prevent misdiagnosis. It would be interesting to test intermediate times between 9 and 24 h of incubation and evaluate the corresponding detection sensitivities. For CP-CRE isolates detected after a 9 h incubation, if MALDI-ToF MS identification could be carried out in the same day, species identification and resistance mechanisms could be obtained within one single day. This technique therefore allows screening of CP-CREs isolated on ESBL agar plates, which are then confirmed by an ICT, as is routinely performed for strains isolated on commercial CP-CRE-selective agars [23]. Finally, this technique also gives a rapid phenotypical detection compared to molecular methods that are expensive and have only a genotypic detection.

However, this approach also had some limitations. Firstly, species identification remains essential in order to prevent misinterpretation. Regarding CP-CRE, confirmation of the type of carbapenemase also remains essential since no difference was noted between isolates expressing the carbapenemase genes *bla_KPC_*, *bla_OXA-48_*, and *bla_IMP-4_* (Appendix A). In the future, it would be interesting to test a higher concentration of ertapenem and/or to include novel antibiotics, e.g., ceftazidime/avibactam and ceftolozane/tazobactam, as proposed by BRUKER^®^ in their MICRONAUT-S Carbapenemases Detection kits [24] to allow these distinctions. Finally, this approach has disadvantages such as the need to include control isolates in each growth monitoring experiment. Indeed, as this method is culture dependent, biases linked to the culture medium may appear. Additionally, we are aware that our collection of non-CRE, non-CP-CRE, and CP-CRE isolates used as controls only reflect the epidemiological situation in New Caledonia. With a view to extending this approach to other settings, it would be useful to examine this technique with a larger number of molecularly typed reference strains from different parts of the world, as suggested by Tamma et al. [20].

### 3.2. Application to Clinical Samples

The active surveillance of CP-CRE in colonization and the rapid detection of colonization of the digestive tract by CP-CRE are very important to limit the selective pressure of carbapenems in patients and to control their spread, notably in healthcare settings. In New Caledonia, several strategies were applied to prevent and control the spread of CP-CRE [8]. Among those, a systematic screening of intensive care unit patients for CP-CRE by rectal swab was performed since the last quarter of 2004. Systematic screening at admission was extended in 2015 to all cross-border patient transfers who are additionally hospitalized in a dedicated unit. Patients at high risk of carriage were screened only at admission; no new swab was performed in the event of a negative result, even when antibiotic treatment had been administered. Up until December 2015, rectal swabs were analysed using only culture methods with ChromID ESBL, followed by synergy test. However, during a rectal swab screening, only one or two colonies are routinely tested. Our results indicate that our method could be complementary to the analyses performed in the diagnostic laboratory by increasing the sampling effort and to limit false-negative diagnostic results. Concerning sampling effort, our method potentially allows the capacity to test a larger number of colonies. Indeed, although no CP-CRE were found in the 30 clinical samples tested here, supplementary resistance phenotypes were detected which may have an important impact on the antibiotic therapy. Regarding, the limitation of false negative diagnostic results, KPC-carbapenemases were detected due to the non-coherence between this ertapenem-supplemented media approach and synergy test. Typically, at a CHT laboratory, based on synergy test only, the result would be the presence of an ESBL, for which no confirmation by a ICT would be performed. In view of these results, the CHT laboratory has now added a temocillin disk (in addition to those listed in the Section 4.2.4) on synergy tests and considered the implementation of this ertapenem-supplemented selective media method for detection of particular β-lactams resistance phenotypes.

### 3.3. First CP-CRE Isolated from Hospital Effluent in New Caledonia

More recently, untreated wastewater has been successfully used for AMR monitoring in human populations [25]. Moreover, One Health approaches through wastewater analysis are now recommended in order to gain a more complete view of antibiotic resistance at a community and ecosystem level [26]. Sewage waters have been shown to accurately reflect the population’s gut microbiota composition, raising the possibility of using wastewater epidemiology to directly gain information about antimicrobial resistance in populations. Often referred to as wastewater-based epidemiology, this technique can give an unbiased insight into the community’s health. Previous work has demonstrated that the supplemented media approach can be effective for detecting CP-CRE in environmental waters [27,28]. Although regarded as laborious and time consuming, culture-based methods allow the detection of some extremely rare resistance types that may go undetected by other methods [29]. In this study, hospital effluents are used to evaluate our method on environmental samples.

Wastewater analysis for antimicrobial resistance is an innovative approach in New Caledonia. Indeed, works focused on AMR were clinical studies, and these results are the first ones to provide antimicrobial resistance data using water analysis. Our method allows the quantitative detection of β-lactams resistance in Enterobacterales in a high bacterial load sample. As reported in other studies [30,31], a majority of presumptive non-CRE were detected in our hospital effluents. In our study, 18 CP-CRE were isolated from hospital effluents. CP-CRE are frequently found in hospital effluent in the world [27,32]. However, this is the first time that such bacteria have been isolated from wastewater in New Caledonia and to our knowledge in PICT’s. Concerning carbapenemase typing, our results are in accordance with the New Caledonian epidemiology. Indeed, since 2013, a significant increase in CP-CRE infections has been described, mostly associated with Enterobacterales harboring an IMP encoded by *bla_IMP-4_* [8]. In our study, 16 CP-CRE harbouring an IMP, among a total 18 CP-CRE revealed in this study, were isolated.

In addition, this study allowed the identification of the first KPC carbapenemases (KPC-2 confirmed by whole genome sequencing) an enzyme never found before in any clinical or environmental sample in New Caledonia. KPC were reported in all Gram-negative members of the “ESKAPE pathogens”, and occur worldwide [33,34]. In the Pacific region, KPC were found in Australia, where they have been involved in outbreaks [35]. The clinical relevance of these enzymes comes from their ability to hydrolyze a broad variety of β-lactams, including carbapenems, cephalosporins and penicillins [36], from their location on conjugative plasmids, and their frequent association with *K. pneumoniae*, an organism notorious for its ability to accumulate and transfer resistance determinants [37]. Revealing this KPC was a very interesting finding because the profile of the synergy test could suggest an ESBL (KPC activity can be partially inhibited by clavulanic acid [38,39]), whereas our method classified it as a presumptive CP-CRE. Finally, ICT confirmed a KPC. Moreover, a *K. oxytoca* harbouring two carbapenemases (IMP and KPC) was found; this is also the first time in New Caledonia that two carbapenemases are found in a same isolate. This result once again supports the interest of this method to avoid missing specific germs at the clinical level. Whole genome sequencing is planned on these isolates to characterize the genes involved and the plasmids harbouring these resistance genes. The specific source and transmission mechanism remains to be determined. One hypothesis could implicate the environment recognized as a reservoir for selection and dissemination of antibiotic-resistant bacteria [40].

## 4. Materials and Methods

### 4.1. Implementation of the Technique

#### 4.1.1. Bacterial Strains Used

Twenty-seven phenotypically characterized β-lactam-resistant Enterobacterales from CHT, and Institut Pasteur de Nouvelle-Calédonie (IPNC) bacterial collections (Table 1) were included. Additionally, two known *K. pneumoniae*, OXA-48 strain 11978 [41] and *K. pneumoniae* KPC from Carbapenemase Producing Enterobacteria French National Reference Centre (Le Kremlin-Bicêtre, France), were also used as controls. Phenotypic β-lactam-resistance mechanisms, antimicrobial susceptibility and carbapenemase class confirmation were performed previously, as described below (Section 4.2.4, Section 4.2.5 and Section 4.2.6). Full characteristics of all the strains are presented in Appendix A.

#### 4.1.2. Growth Dynamics over Time in Presence of Ertapenem

Growth in the presence of ertapenem was assessed in aerobic conditions in microplate wells (96 wells) in Lysogeny Broth medium (LB, tryptone: 10 g·L^−1^, yeast extract 5 g·L^−1^, NaCl: 10 g·L^−1^) supplemented with two different concentrations of ertapenem. Wells (1 per concentration) were filled with 50 μL of 4X ertapenem solution and 150 μL of LB. No-ertapenem control wells were filled with 50 µL of sterile distilled water and 150 µL of LB. According to the breakpoint value of MIC for ertapenem [15] and MIC obtained using Broth Microdilution method (Table 1), two concentrations were tested (0.5 µg·mL^−1^ and 4 µg·mL^−1^) together with control (0 µg·mL^−1^). Using a sterile toothpick, isolated colonies from ChromID ESBL agar plates were sampled and inoculated into the three corresponding wells. For clinical or environmental samples, control isolates were systematically added (CP-CRE, non-CP-CRE and non-CRE) in order to control for variability among experimentations. The microplate was incubated at 37 °C under 100 r·min^−1^ orbital shaking. Cell growth was monitored by measuring the A_620nm_ with a microplate spectrophotometer (Multiskan™ FC, Thermo Fisher Scientific, Waltham, MA, USA) every three hours for 9 h, then a final measurement was carried out at 24 h. Finally, a subculture (37 °C; overnight) of each well was inoculated on ChromID ESBL agar plates to check viability and purity.

### 4.2. Application on Clinical and Environmental Samples

#### 4.2.1. Samples Collection

Clinical samples corresponded to 30 anonymous ChromID ESBL agar plates from Highly Resistant Bacteria (HRB) systematic screening performed at CHT laboratory and recovered during two weeks in February 2022. Environmental samples were raw hospital wastewater effluents from CHT, collected before being pumped to a wastewater treatment plant (December 2021–June 2022). Briefly, 24 h composite sampling used an automatic sampler (ISCO 6712, TELEDYNE ISCO, Lincoln, NE, USA) which recovered 500 mL every 30 min. After 24 h, composite samples were gathered, and 500 mL was collected in a sterile bottle, placed on ice and transported to the laboratory within 1 h for testing. Appropriate dilutions (in order to have isolated colonies) were spread on ChromID ESBL agar plates (bioMérieux, Marcy-l’Étoile, France) and incubated at 37 °C overnight.

#### 4.2.2. Growth in Presence of Ertapenem

According to the number of colonies and phenotypes observed on each ChromID ESBL agar plates, 20–30 (for environmental samples) or 3–16 (for clinical samples) isolated colonies were assessed per sample as described in Section 4.1.2. The number of isolates for each presumptive resistance mechanism was recorded, but only presumptive CP-CRE were kept for species identification, synergy tests, antimicrobial Susceptibility testing, and carbapenemase class confirmation.

#### 4.2.3. Species Identification

A rapid identification of the microorganisms based upon protein profiles was performed on the MALDI Biotyper system with MALDI-ToF Mass Spectrometry (Infectio MALDI, microflexLT-MS, Bruker, Billerica, MA, USA). Selected bacterial colonies were directly placed on the MALDI-ToF target (MSP 96 target polish steel BC, microscout Target), followed by addition of CHCA matrix solution (1 µL α-cyano-4-hydroxycinnaminic acid 50% acetonitrile–2.5% trifluoro acetic acid). It was then dried at room temperature to allow co-crystallisation. The plates were then subjected to MALDI-ToF MS measurement. The mass spectra obtained from each sample were imported into the Bio-Typer software (Bruker daltonics) and analyzed by standard pattern matching. Results of the pattern-matching were expressed with scores ranging from 0 to 3. Scores below 1.7 were regarded as unreliable identification; a score of ≥1.7 was regarded as identification of genus, and a score of ≥2 indicated species identification as recommended by the manufacturer.

#### 4.2.4. Synergy Test

Detection of carbapenemase production was performed using ESBL + AmpC Screen Kit (Rosco Diagnostica, Taastrup, Danmark) consisting of cefotaxim 30 µg (CTX) and the corresponding combination disk with clavulanic acid (CTX+C), cloxacillin (CTXCX), and clavulanic acid and cloxacillin (CTXCC). In addition, Bio-Rad antimicrobial susceptibility disks of cefepime 30 µg (FEP), 10 µg ertapenem (ETP), and 30 µg ceftolozane/10 µg tazobactam combination (CLT) were used. Disks were placed on Mueller Hinton agar plates (Oxoid, Dardilly, France) inoculated with 0.5 McFarland bacterial suspensions in 0.85% saline water. An increase in zone diameter of ≥5 mm with the addition of clavulanic acid compared to cefotaxim and associated with a sensibility to ertapenem and resistance to cefepime was interpreted as a positive result for ESBL. An increase in zone diameter of ≥5 mm with the addition of cloxacillin compared to the cefotaxime, associated most of the time with sensibility to cefepime and resistance to ertapenem, was interpreted as a positive result for an AmpC β-lactamase. Recuperation only by the two inhibitors suggested the presence of EBSL and AmpC β-lactamase. No recuperation by the two inhibitors (separately or together) suggested presence of a carbapenemase [42]. Moreover, contact with ceftolozane/tazobactam disk also suggested the presence of an IMP carbapenemase [43]. CP-CRE isolates were kept for carbapenemase class confirmation using ICT.

#### 4.2.5. Antimicrobial Susceptibility Testing

Antibiotic susceptibility profiles were obtained using the VITEK^®^ 2 system (bioMérieux, Marcy l’Etoile, France) [44] and interpreted according to the European Committee on Antimicrobial Susceptibility Testing [15]. In VITEK^®^ 2 system, MIC is determined by comparing the growth of the patient isolate to the growth of isolates with known MICs. This method is equivalent to having a standard curve recorded in VITEK^®^ 2, which is able to correlate the reference MICs to the activity of the organisms in the antibiotic containing wells. Only three concentrations are tested, allowing six different MICs to be determined.

#### 4.2.6. Carbapenemase Class Confirmation

Carbapenemase class confirmation was performed by NG-Test^®^ CARBA-5 (NG BIOTECH Laboratories, Guipry, France) [45] or O.K.N.V.I. RESIST-5 (CORIS BioConcept, Gembloux, Belgique) [46] following the manufacturer’s instructions. These kits allow the confirmation of the occurrence of NDM-, IMP-, VIM-, OXA48- and KPC-type carbapenemases. Finally, confirmed CP-CRE strains were stored at −80 °C.

### 4.3. Growth over Time and Carbapenemase Class Confirmation after Nine Hours of Incubation

In order to improve turnaround time for diagnostic purposes, growth in the presence of ertapenem was carried out under the same conditions as described above (2.2) on 64 colonies isolated from another environmental sample (MED7; 24 h composite as described above, collected in November 2022). However, after 9 h of incubation, an ICT was used on presumptive CP-CRE in wells at 4 µg·mL^−1^ of ertapenem and in wells with LB only (to avoid bacteriostatic effect observed at 4 µg·mL^−1^ of ertapenem).

### 4.4. Statistical Analysis

All statistical analyses were performed using R-Studio software version 1.3.1093 [47]. All data were analysed with nonparametric tests (Kruskal-Wallis test, followed by a Fisher least significant difference (LSD) test, with *p* < 0.05 considered as a significant difference), since data did not follow the assumptions for a parametric test.

### 4.5. Ethical Statement

Eight IMP-CP-CRE isolated from clinical samples were used as positive controls, 30 anonymous ESBL plates from HRB systemic screening (patients’ rectal swabs), and microbiological analysis results associated were provided by CHT laboratory. Before being transmitted, all patients’ data were removed, and all samples and isolates were handled anonymously in agreement with the requirements of the Public Health Authority. No further ethical clearance was required as per local regulations. Nevertheless, a Material Transfer Agreement was signed between CHT and IPNC.

## 5. Conclusions

Here, an easy, practical, and cost-effective method is proposed to have a result in one day, which would help clinicians rapidly identify patients carrying CP-CRE for better individual and collective management. Application of this method allowed to test a high number of colonies from a high bacterial load sample, and has revealed the presence of CP-CRE in all hospital effluent samples studied with occurrence of IMP and KPC type carbapenemases. Moreover, our results have shown that majority of CP-CRE isolated are resistant to other antibiotics. Thereafter, genomic analyses will make it possible to compare the isolates found in hospital effluents with clinical isolates from patients in New Caledonia. Moreover, these analyses will be extended to other sites in New Caledonia (inlet of wastewater treatment plant, swimming area) in order to draw up an inventory of the carbapenemases circulating in New Caledonia and to develop an AMR One Health approach in the Pacific region.

## Figures and Tables

**Figure 1 antibiotics-12-00392-f001:**
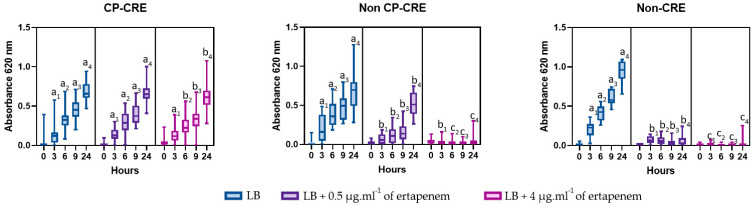
Growth over time according to ertapenem concentration (0, 0.5 or 4 µg·mL^−1^) for carbapenemase-producing-carbapenem-resistant-Enterobacterales (CP-CRE; *n* = 57), non-carbapenemase-producing-carbapenem-resistant-Enterobacterales (non-CP-CRE; *n* = 23) and non-carbapenem-resistant Enterobacterales (Non-CRE; *n* = 11). The same underscript number to a letter reflects time-point specific analyses. Whiskers correspond to min and max values. Different letters above bars indicate significant differences between ertapenem concentration as determined by Fisher’s LSD test (*p* < 0.05).

**Figure 2 antibiotics-12-00392-f002:**
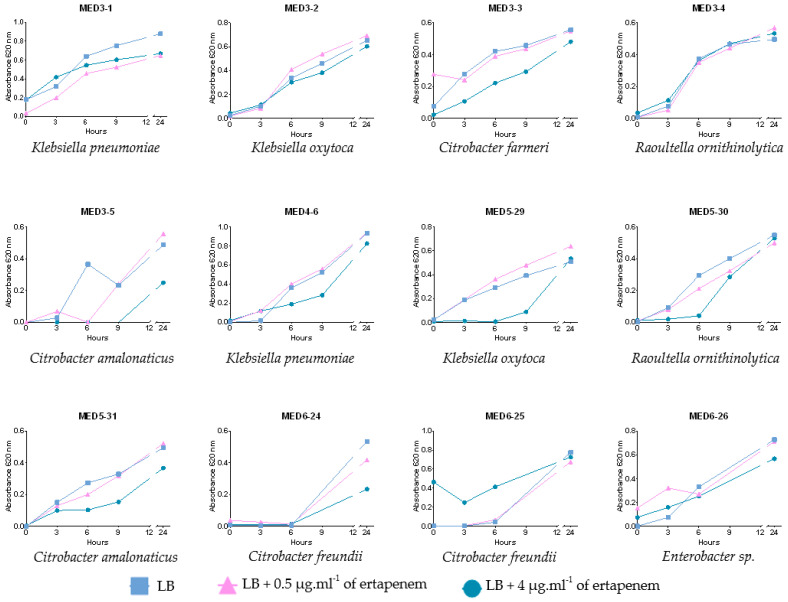
Growth over time according to the ertapenem concentration (0, 0.5 or 4 µg·mL^−1^) and species identification of presumptive carbapenemase-producing-carbapenem-resistant-Enterobacterales isolated from MED3, MED4, MED5, and MED6.

**Figure 3 antibiotics-12-00392-f003:**
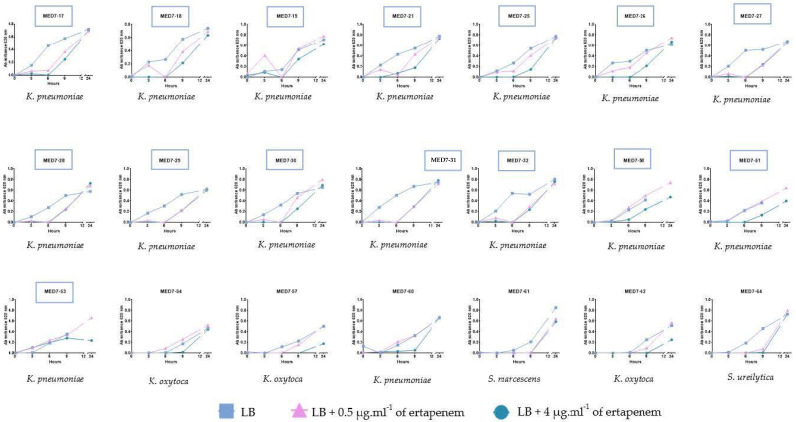
Growth over time according to the ertapenem concentration (0, 0.5 or 4 µg·mL^−1^) and species identification of presumptive carbapenemase-producing carbapenem-resistant-Enterobacterales isolated from MED7. Presumptive CP-CRE after nine hours of incubation are framed.

**Table 1 antibiotics-12-00392-t001:** Characteristics of β-lactam-resistant Enterobacterales used as controls in this study.

Name	MALDI-ToF MS Identification	Phenotypic β-Lactam-Resistant Mechanisms ^a^	Carbapenemase Class ^b^	Ertapenem Resistance ^c^		Presumptive Carbapenem Resistance Mechanism
					MIC µg·mL^−1^	
8210610788	*Citrobacter freundii*	Carbapenemase	IMP	I	2	CP-CRE
8211020125	*Citrobacter freundii*	Carbapenemase	IMP	S	≤0.5	CP-CRE
8211020127	*Citrobacter freundii*	Carbapenemase	IMP	R	>4	CP-CRE
8211320250	*Citrobacter freundii*	Carbapenemase	IMP	R	4	CP-CRE
8211510181	*Enterobacter* sp.	Carbapenemase	IMP	R	>4	CP-CRE
8211760544	*Escherichia coli*	Carbapenemase	IMP	S	≤0.5	CP-CRE
8213210685	*Klebsiella pneumoniae*	Carbapenemase	IMP	R	4	CP-CRE
8211480320	*Escherichia coli*	Carbapenemase	IMP	R	4	CP-CRE
CNR1	*Klebsiella pneumoniae*	Carbapenemase	KPC	R	>4	CP-CRE
CNR2 (ATCC 11978)	*Klebsiella pneumoniae*	Carbapenemase	OXA-48	R	>4	CP-CRE
AH	*Enterobacter cloacae*	Carbapenemase	IMP	R	>4	CP-CRE
AI	*Citrobacter freundii*	Carbapenemase	IMP	R	4	CP-CRE
AM	*Klebsiella pneumoniae*	Carbapenemase	IMP	R	2	CP-CRE
BP	*Klebsiella pneumoniae*	Carbapenemase	IMP	R	2	CP-CRE
LHE 358.102	*Enterobacter* sp.	Carbapenemase	IMP	R	4	CP-CRE
LHE 358.157	*Enterobacter* sp.	Carbapenemase	IMP	R	4	CP-CRE
BF	*Enterobacter* sp.	AmpC	-	R	2	Non-CP-CRE
BM	*Escherichia coli*	AmpC	-	R	2	Non-CP-CRE
BS	*Klebsiella pneumoniae*	ESBL and AmpC	-	R	4	Non-CP-CRE
BR	*Klebsiella aerogenes*	AmpC	-	S	≤0.5	Non-CP-CRE
AY	*Enterobacter* sp.	ESBL and AmpC	-	R	4	Non-CP-CRE
BK	*Escherichia coli*	ESBL	-	S	≤0.5	Non-CRE
BQ	*Enterobacter* sp.	ESBL	-	S	≤0.5	Non-CRE
AD	*Klebsiella pneumoniae*	ESBL	-	S	≤0.5	Non-CRE
AR	*Klebsiella pneumoniae*	ESBL	-	S	≤0.5	Non-CRE
BO	*Serratia marcescens*	ESBL	-	S	≤0.5	Non-CRE
BI	*Klebsiella pneumoniae*	ESBL	-	S	≤0.5	Non-CRE

^a^ Determined using synergy test, ^b^ Confirmed using NG-Test^®^ CARBA-5 (IMP: IMP-type carbapenemase; KPC: KPC type carbapenemase, OXA-48-type carbapenemase), ^c^ Determined using VITEK^®^ 2 system.

**Table 2 antibiotics-12-00392-t002:** Comparison of results obtained in Territorial Hospital Center laboratory with ertapenem culture approach.

Sample	Ertapenem-Supplemented Selective Media	Territorial Hospital Center Laboratory		Correspondence
	Presumptive Resistance Mechanisms	MALDI-ToF MS Identification	Type of β-Lactamase ^a^	
CS-1	Non-CRE (*n*= 3)	*E. coli*	ESBL	Yes
		*K. pneumoniae*	-	
CS-2	Non-CP-CRE (*n*= 3)	*Enterobacter* sp.	AmpC	Yes
CS-3	Non-CRE (*n*= 3)	*E. coli*	ESBL	Yes
		*K. pneumoniae*	ESBL	
		*C. freundii*	ESBL	
CS-4	Non-CP-CRE (*n*= 3)	*Enterobacter* sp.	AmpC	Yes
CS-5	Non-CP-CRE (*n*= 3)	*Enterobacter* sp.	AmpC	Yes
CS-6	2 Non-CRE and 1 Non-CP-CRE	*K. pneumoniae*	ESBL	Yes
		*E. cloacae*	AmpC	
CS-7	Non-CRE (*n*= 5)	*K. pneumoniae*	ESBL	Yes
CS-8	Non-CRE (*n*= 5)	*K. pneumoniae*	ESBL	Yes
CS-9	1 Non-CRE (red colony) and 4 Non-CP-CRE (green colonies)	*E. coli*	ESBL	Yes
		*K. aerogenes*	AmpC	
CS-10	Non-CRE (*n*= 5)	*K. pneumoniae*	ESBL	Yes
CS-11	Non-CRE (*n*= 5)	*Enterobacter sp.*	ESBL	Yes
CS-12	Non-CRE (*n*= 3)	*K. pneumoniae*	ESBL	Yes
CS-13	Non-CRE (*n*= 5)	*Enterobacter* sp.	AmpC	No
CS-14	2 Non-CRE and 1 Non-CP-CRE	*Enterobacter* sp.	AmpC	Supplementary phenotype
CS-15	Non-CRE (*n*= 5)	*Enterobacter* sp.	AmpC	No
CS-16	Non-CRE (*n*= 3)	*K. pneumoniae*	ESBL	Yes
CS-17	Non-CRE (*n*= 3)	*S. marcescens*	ESBL	Yes
CS-18	Non-CRE (*n*= 3)	*K. pneumoniae*	ESBL	Yes
CS-19	Non-CRE (*n*= 3)	*K. pneumoniae*	ESBL	Yes
CS-20	Non-CRE (*n*= 6)	*E. coli*	ESBL	Yes
CS-21	Non-CRE (*n*= 3)	*K. aerogenes*	AmpC	No
CS-22	Non-CP-CRE (*n*= 8)	*Enterobacter* sp.	AmpC	Yes
CS-23	Non-CRE (*n*= 12)	*K. pneumoniae*	ESBL	Yes
CS-24	Non-CRE (*n*= 8)	*K. pneumoniae*	ESBL	Yes
CS-25	2 Non-CRE and 2 Non-CP-CRE	*Enterobacter* sp.	ESBL + AmpC	Yes
CS-26	Non-CRE (*n*= 16)	*E. coli*	ESBL	Yes
CS-27	Non-CRE (*n*= 3)	*C. freundii*	AmpC but sensitive to ertapenem	Yes
CS-28	Non-CRE (*n*= 16)	*Enterobacter* sp.	ESBL	Yes
CS-29	Non-CP-CRE (*n*= 8)	*Enterobacter* sp.	AmpC	Yes
CS-30	Non-CP-CRE (*n*= 16)	*Enterobacter* sp.	AmpC	Yes

^a^ Determined using synergy test.

**Table 3 antibiotics-12-00392-t003:** Presumptive CP-CRE, non-CP-CRE, and non-CRE isolated from wastewater samples after 24 h of incubation.

Samples	Colony Number Tested	Presumptive CP-CRE ^a^	Presumptive Non-CP-CRE ^a^	Presumptive Non-CRE ^a^
MED3	24	5 (21%)	8 (33%)	11 (46%)
MED4	23	1 (4.5%)	1 (4.5%)	21 (91%)
MED5	31	3 (10%)	0	28 (90%)
MED6	26	3 (11%)	2 (8%)	21 (81%)
MED7	64	21 (33%)	6 (9%)	37 (58%)

^a^ The percentages correspond to the number of colonies for each resistance mechanism in relation to the number of colonies tested.

**Table 4 antibiotics-12-00392-t004:** Species, type of β-lactamase, ertapenem MIC, and carbapenemase’s class of presumptive CP-CRE isolated from wastewater samples.

Sample	MALDI-ToF MS Identification	Type of β-Lactamase ^a^	Class of Carbapenemase ^b^	Antimicrobial Susceptibility ^c^																			
				Ampicillin	Amoxicillin + Clavulanic Acid (Urine + Other)	Ticarcillin	Ticarcillin + Clavulanic Acid	Piperacillin	Piperacillin + Tazobactam	Cefotaxim	Ceftazidim	Cefoxitin	Imipenem	Ertapenem	Ertapenem MIC µg·mL^−1^	Amikacin	Gentamicin	Nalidixic Acid	Ofloxacin	Ciprofloxacin	Levofloxacin	Nitrofurantoin	Trime Thoprim/Sulfamethoxazol
MED3-1	*K. pneumoniae*	Carbapenemase	IMP	R	R	R	R	R	R	R	R	R	R	R	4	I	R	R	R	R	R	I	R
MED3-2	*K. oxytoca*	Carbapenemase	IMP	R	R	R	R	R	R	R	R	R	I	R	2	S	R	S	I	S	S	S	S
MED3-3	*C. farmeri*	Carbapenemase	IMP	R	R	R	R	-	R	R	R	R	R	R	2	I	R	R	R	R	-	S	R
MED3-4	*R. ornithinolytica*	Carbapenemase	IMP	R	R	R	R	-	I	R	R	R	R	R	4	I	R	R	R	I	-	S	S
MED3-5	*C. amalonaticus*	ESBL	KPC	R	R	R	R	-	R	R	I	S	R	R	2	R	R	-	R	R	-	S	S
MED4-6	*K. pneumoniae*	Carbapenemase	IMP	R	R	R	R	R	R	R	R	R	R	R	4	I	R	R	R	S	-	R	R
MED5-29	*K. oxytoca*	Carbapenemase	IMP	R	R	R	R	R	R	R	R	R	I	R	4	S	R	-	R	I	-	S	R
MED5-30	*R. ornithinolytica*	Carbapenemase	IMP	R	R	R	R	-	I	R	R	R	R	R	4	S	R	-	I	S	-	I	R
MED5-31	*C. amalonaticus*	ESBL	KPC	R	R	R	R	-	R	R	I	S	R	R	4	R	R	-	R	R	-	S	S
MED6-24	*C. freundii*	Carbapenemase	IMP	R	R	R	R	R	R	R	R	R	R	R	4	R	R	-	R	R	R	S	R
MED6-25	*C. freundii*	Carbapenemase	IMP	R	R	R	R	R	R	R	R	R	R	R	4	S	R	-	R	R	R	S	R
MED6-26	*Enterobacter* sp.	Carbapenemase	IMP	R	R	R	R	R	R	R	R	R	R	R	4	S	R	-	R	S	-	I	S
MED7-18	*K. pneumoniae*	Carbapenemase	IMP	R	R	R	R	R	R	R	R	R	R	R	>4	S	R	R	R	R	R	R	S
MED7-21	*K. pneumoniae*	Carbapenemase	IMP	R	R	R	R	R	R	R	R	R	R	R	>4	S	R	R	R	R	R	R	S
MED7-26	*K. pneumoniae*	Carbapenemase	IMP	R	R	R	R	R	R	R	R	R	R	R	>4	S	R	R	R	R	R	I	S
MED7-54	*K. oxytoca*	Carbapenemase	IMP + KPC	R	R	R	R	R	R	R	R	R	R	R	>4	S	R	R	R	R	R	S	R
MED7-57	*K. oxytoca*	Carbapenemase	IMP	R	R	R	R	R	R	R	R	R	R	R	>4	R	R	R	R	R	R	S	R
MED7-61	*S. marcescens*	Carbapenemase	IMP	R	R	R	R	R	R	R	R	R	R	R	>4	R	R	R	R	S	R	R	R

^a^ Determined using synergy test, ^b^ Confirmed using NG-Test^®^ CARBA-5 or O.K.N.V.I. RESIST-5 (IMP: IMP-type carbapenemase; KPC: KPC type carbapenemase), ^c^ Determined using VITEK^®^2 system (R: Resistant, I: Intermediate, S: Susceptible, -: not tested.

**Table 5 antibiotics-12-00392-t005:** CP-CRE from MED7 sample where Tn stands for n hours of incubation.

Name	Species	Presumptive CP-CRE at T9	Immunochromatography Test (ICT) on LB + 4 µg·mL^−1^ Wells at T9	ICT on LB Wells at T9	Presumptive CP-CRE at T24	ICT on LB + 4 µg·mL^−1^ Wells at T24	Type of β-Lactamase *	ICT ^#^
MED7-17 ^Ϯ^	*K. pneumoniae*	Yes	-	-	Yes	IMP	-	-
MED7-18 ^Ϯ^	*K. pneumoniae*	Yes	No band	-	Yes	IMP	Carbapenemase	-
MED7-19 ^Ϯ^	*K. pneumoniae*	Yes	-	-	Yes	IMP	-	-
MED7-21 ^Ϯ^	*K. pneumoniae*	Yes	IMP	-	Yes	IMP	Carbapenemase	-
MED7-25 ^Ϯ^	*K. pneumoniae*	Yes	IMP	-	Yes	IMP	-	-
MED7-26 ^Ϯ^	*K. pneumoniae*	Yes	No band	-	Yes	IMP	Carbapenemase	-
MED7-27 ^Ϯ^	*K. pneumoniae*	Yes	IMP	-	Yes	IMP	-	-
MED7-28 ^Ϯ^	*K. pneumoniae*	Yes	IMP	-	Yes	IMP	-	-
MED7-29 ^Ϯ^	*K. pneumoniae*	Yes	IMP	-	Yes	IMP	-	-
MED7-30 ^Ϯ^	*K. pneumoniae*	Yes	IMP	-	Yes	IMP	-	-
MED7-31 ^Ϯ^	*K. pneumoniae*	Yes	IMP	-	Yes	IMP	-	-
MED7-32 ^Ϯ^	*K. pneumoniae*	Yes	IMP	-	Yes	IMP	-	-
MED7-50 ^Ϯ^	*K. pneumoniae*	Yes	No band	IMP	Yes	-	-	-
MED7-51 ^Ϯ^	*K. pneumoniae*	Yes	No band	No band	Yes	-	Carbapenemase	IMP
MED7-53 ^Ϯ^	*K. pneumoniae*	Yes	IMP	IMP	Yes	-	-	-
MED7-54	*K. oxytoca*	No	-	-	Yes	-	Carbapenemase	IMP + KPC
MED7-57	*K. oxytoca*	No	-	-	Yes	-	Carbapenemase	IMP
MED7-60	*K. pneumoniae*	No	-	-	Yes	-	Carbapenemase	IMP
MED7-61	*S. marcescens*	No	-	-	Yes	-	Carbapenemase	IMP
MED7-62	*K. oxytoca*	No	-	-	Yes	-	Carbapenemase	IMP
MED7-64	*S. ureilytica*	No	-	-	Yes	-	Carbapenemase	IMP
		(15/21) 71%	(10/13) 77%		(21/21) 100%	(12/12) 100%		

^Ϯ^ clone suspicion, * determined using synergy test, ^#^ performed after synergy test using using NG-Test^®^ CARBA-5 or O.K.N.V.I. RESIST-5 (IMP: IMP-type carbapenemase; KPC: KPC type carbapenemase), -: not tested.

## Data Availability

Not applicable.

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
