# Peer review of "Ertapenem Supplemented Selective Media as a New Strategy to Distinguish β-Lactam-Resistant Enterobacterales: Application to Clinical and Wastewater Samples"

_antibiotics, 2023, doi:10.3390/antibiotics12020392_

Round 1
Reviewer 1 Report
This manuscript by Bourles al. reports the simple method to classify CP-CRE, Non-CP-CRE, and Non-CRE on the basis of bacterial growth in LB broth containing ertapenem. This method is easy, practical, and cost-effective. The method would be a great help to characterize antibiotic resistant bacteria in clinical microbiology laboratories. This reviewer asks the authors to consider the following minor points.
Lines 113-115: IMP-producing strains 8211020125 and 8211760544 are categorized as CP-CRE despite these two are susceptible to ertapenem (MIC, below 0.5 ug/ml). It was suitable that these two were determined to be Non-CRE. In addition, the BR strain which is susceptible to ertapenem was categorized as Non-CP-CRE (actually Non-CRE). On the basis of ertapenem susceptibility results determined by VITEK-2 system, the author’s methods show some disagreements. Thus, the phrase “A perfect match was found…” should be avoided. Such disagreement should be commented elsewhere as a limitation for the author’s method.
Line 363: The first identification of KPC is significant. The authors should at least determine the genotype of KPC (KPC-2?, KPC-3?…).
Table 1 and others. “Serratia” is now categorized as Yersiniaceae (not Enterobacteriaecea). Enterobacteriaceae should be Enterobacterales in some parts in the manuscript.
Table 2 and 5. The MIC values of ertapenem for each strain should be added.
Figure 3 and 4. The usage guides “LB + 0.5” and “LB + 4” should be “LB + 0.5 ug.ml-1 of ertapenem” and “LB + 4 ug.ml-1 of ertapenem”, respectively.
Author Response
Response to Reviewer 1 Comments
This manuscript by Bourles al. reports the simple method to classify CP-CRE, Non-CP-CRE, and Non-CRE on the basis of bacterial growth in LB broth containing ertapenem. This method is easy, practical, and cost-effective. The method would be a great help to characterize antibiotic resistant bacteria in clinical microbiology laboratories. This reviewer asks the authors to consider the following minor points
Lines 113-115: IMP-producing strains 8211020125 and 8211760544 are categorized as CP-CRE despite these two are susceptible to ertapenem (MIC, below 0.5 ug/ml). It was suitable that these two were determined to be Non-CRE. In addition, the BR strain which is susceptible to ertapenem was categorized as Non-CP-CRE (actually Non-CRE). On the basis of ertapenem susceptibility results determined by VITEK-2 system, the author’s methods show some disagreements. Thus, the phrase “A perfect match was found…” should be avoided. Such disagreement should be commented elsewhere as a limitation for the author’s method.
Response: Indeed, this is an important point. IMPs have the characteristic to hydrolyze ertapenem weakly, this is what is shown here. We have indeed the presence of a carbapenemase (confirmed using synergy test and immunochromatographic test) while the MIC obtained using VITEK-2 system are below 0.5 µg.ml-1. This is an additional argument supporting our method. Results of VITEK-2 system are discussed in the revised MS, so from our point of view we can retain the "A perfect match was found...".
Line 363: The first identification of KPC is significant. The authors should at least determine the genotype of KPC (KPC-2?, KPC-3?…).
Response. For the KPC, it is KPC-2.((confirmed by whole genome sequencing), information is added the revised MS (Line 388).
Table 1 and others. “Serratia” is now categorized as Yersiniaceae (not Enterobacteriaecea). Enterobacteriaceae should be Enterobacterales in some parts in the manuscript.
Response: Ok we use Enterobacterales in the revised MS.
Table 2 and 5. The MIC values of ertapenem for each strain should be added.
Response: Unfortunately, we do not have this information for the strains and are no longer able to recover them (isolates have been discarded).
Figure 3 and 4. The usage guides “LB + 0.5” and “LB + 4” should be “LB + 0.5 ug.ml-1 of ertapenem” and “LB + 4 ug.ml-1 of ertapenem”, respectively.
Response: We agree with this comment and modified the revised MS accordingly.
Reviewer 2 Report
Title: Ertapenem supplemented selective media to distinguish beta-lactams resistant Enterobacteri-aceae: application to clinical and wastewater samples
This manuscript is not ready to publish in the journal as many weak points were presented in it. However, I do believe that if they can improve the manuscripts following all comments. It might have a chance to publish in the journal.
Comments
1. Topic: Please rewrite
I have suggested
"Ertapenem supplemented selective media as a new strategy to distinguish beta-lactams resistant Enterobacteriaceae: application to clinical and wastewater samples"
Abstract
2. Please use part tense in the objective of this study. It would be better if the author use passive voice instead of we propose ……
3. The sentence “In New Caledonia the majority of carbapenemamases” It need a comma “In New Caledonia, the majority of carbapenemamases”
4. It would be better if the authors explain the results by the values.
5. This sentence “Finally, this cost-effective method, in addition to providing fast results, allows a set of controls to avoid diagnostic errors.” It should be rewritten.
6. This sentence should be rewritten “All of these elements argue that this method could be applied in the Pacific islands or other resource-limited settings, where very little data exists on the β-lactam resistance of Enterobacteriaceae Pacific islands or other resource-limited settings” The authors should explain as this is the new method or strategy to distinguish beta-lactams resistant Enterobacteriaceae…..
Introduction
7. Line 41-43, please mention the bacteria in Enterobacteriaceae
8. Please include the objectives of the study at the end of the introduction.
Methods
9. Line 384, please check the space of the word “4.1. . Implementation of the technique”
10. Line 386, the sentence “We included 27 phenotypically characterized β-lactam-resistant Enterobacteriaceae” please, rewrite using passive voice.
11. Please include the reference of each methodology.
12. Line 417, December 2021-June 2022.
13. Line 445-447, please include the brand of the antibiotic discs.
14. Please use one space between the number and the letter. For example, 5 mm. However, most of the sentences was written as 5mm.
15. Line 461, please include the principle of Antibiotic susceptibility using the VITEK-2 system.
16. Line 471, 478, 483, please check the space; “4.3. . Growth over time and Carbapenemase class confirmation after a 9-hour incubation” “4.4. . Statistical analysis” “4.5. . Ethical statement”
Results
17. The authors presented a lot of the results. However, some of them are raw data. So, it is better if the authors summarize the data, and present them.
18. For the results, the authors should present the summarize the data or representative results. The other data can be presented as supplementary data.
19. Line 208, please check the bold letter.
20. Please use the figures with high resolution
Discussion
21. All the results; please describe and discuss the compact and key the results.
References.
22. There are many references. In general, there are 30-40 references for each manuscript (research articles). Please delete some unnecessary references.
Author Response
Response to Reviewer 2 Comments
This manuscript is not ready to publish in the journal as many weak points were presented in it. However, I do believe that if they can improve the manuscripts following all comments. It might have a chance to publish in the journal.
- Topic: Please rewrite
I have suggested
"Ertapenem supplemented selective media as a new strategy to distinguish beta-lactams resistant Enterobacteriaceae: application to clinical and wastewater samples"
We thank the reviewer for this suggested phrasing and have modified the revised MS accordingly.
Abstract
- Please use part tense in the objective of this study. It would be better if the author use passive voice instead of we propose ……
We have modified the revised MS to take this suggestion into account (Lines 19, 22, 82, 84, 89, 95, 222, 372, 378, 384).
- The sentence “In New Caledonia the majority of carbapenemamases” It need a comma “In New Caledonia, the majority of carbapenemamases”
We have modified as suggested (Line 17).
- It would be better if the authors explain the results by the values.
In our opinion it is not ideal to include values in the abstract.
- This sentence “Finally, this cost-effective method, in addition to providing fast results, allows a set of controls to avoid diagnostic errors.” It should be rewritten.
We have rephrased the end of the abstract to take this suggestion into account (Lines 29-33).
- This sentence should be rewritten “All of these elements argue that this method could be applied in the Pacific islands or other resource-limited settings, where very little data exists on the β-lactam resistance of Enterobacteriaceae Pacific islands or other resource-limited settings” The authors should explain as this is the new method or strategy to distinguish beta-lactams resistant Enterobacteriaceae…..
We have rephrased this sentence to take this suggestion into account (Lines 29-33).
Introduction
- Line 41-43, please mention the bacteria in Enterobacteriaceae
We have modified in the revised MS to take this suggestion into account (Line 46).
- Please include the objectives of the study at the end of the introduction.
We have done our best to make the objective clearer in our revised version of the MS (Line 84).
Methods
- Line 384, please check the space of the word “4.1. . Implementation of the technique”
Done (Line 405)
- Line 386, the sentence “We included 27 phenotypically characterized β-lactam-resistant Enterobacteriaceae” please, rewrite using passive voice.
We have rephrased to use passive voice as suggested (Line 407).
- Please include the reference of each methodolo:gy.
Reference for VITEK-2 system was included (Line 471).
- Line 417, December 2021-June 2022.
We have modified as suggested (Line 433).
- Line 445-447, please include the brand of the antibiotic discs.
Done (Line 457).
- Please use one space between the number and the letter.For example, 5 mm. However, most of the sentences was written as 5mm.
We have modified as suggested in revised MS.
- Line 461, please include the principle of Antibiotic susceptibility using the VITEK-2 system.
The revised MS now includes this principle (Lines 472-475)
- Line 471, 478, 483, please check the space; “4.3. . Growth over time and Carbapenemase class confirmation after a 9-hour incubation” “4.4. . Statistical analysis” “4.5. . Ethical statement”
Done
Results
- The authors presented a lot of the results. However, some of them are raw data. So, it is better if the authors summarize the data, and present them.
- For the results, the authors should present the summarize the data or representative results. The other data can be presented as supplementary data.
We now propose to move Figure 2 to a supplementary Figure and to delete Figure 1b.
- Line 208, please check the bold letter.
Done
- Please use the figures with high resolution
Done
Discussion
- All the results; please describe and discuss the compact and key the results.
We have taken this into account and propose new formulations in the revised MS.
References.
- There are many references. In general, there are 30-40 references for each manuscript (research articles). Please delete some unnecessary references.
Some non-essential references were removed: there are now 47 references in the revised manuscript.
Reviewer 3 Report
The manuscript entitled "Ertapenem supplemented selective media to distinguish beta-lactam resistant Enterobacteriaceae: application to clinical and wastewater samples" publishes the results of a study on the resistance of enterobacteria to antibiotics. The materials are collected in New Caledonia and are of exceptional interest. The study was carried out competently, the methods were selected adequately, the results are presented clearly. The text of the article is logical, the narrative is consistent, the conclusions are confirmed by the results. The conclusions should have been spelled out more clearly. A small correction of the English language is required. It is recommended to accept the article after minor changes.
1) Lines 104, 125, 224, 311, 331, 384, 411, 471, 478 and 483. There is an unnecessary dot at the beginning of the line.
2) Pages 3, 5 and 8. There is too much free space at the end of the page.
3) Line 88. It is advisable to remove the hyperlink from the text by placing it in the list of literary sources.
4) Line 501. The dot at the end of the sentence is missing.
5) Line 62. It is advisable to replace the not quite appropriate word "captive". In general, the description of the island archipelago should be more related to the topic of the article.
6) Chapter 4.1.2. With the help of a multiscan, measurements can be carried out every hour. Why was such a scheme chosen – once every 3 hours and at the end, after 24 hours? It would be possible to get a more detailed picture, almost without additional effort, simply by adding a protocol.
7) Chapter 4.1.2. 96 holes would allow more concentrations to be used. Why did the authors choose only two concentrations?
8) Figures 1 and 2 could be drawn not in the form of roadblocks, but in the form of curved lines, and three lines can be placed together on one graph. Or more than three if the authors took more than two concentrations. It would be more visual.
9) The conclusions of the entire study should be written more specifically. In chapter 2, the conclusions are formulated too concretely, in chapter 3, basically, the situation as a whole is described. It is advisable to add to chapter 3 a few generalizing suggestions concerning only the work of the authors. Chapter 5 should be rewritten, leaving in it specific conclusions from the study.
Author Response
Response to Reviewer 3 comments
The manuscript entitled "Ertapenem supplemented selective media to distinguish beta-lactam resistant Enterobacteriaceae: application to clinical and wastewater samples" publishes the results of a study on the resistance of enterobacteria to antibiotics. The materials are collected in New Caledonia and are of exceptional interest. The study was carried out competently, the methods were selected adequately, the results are presented clearly. The text of the article is logical, the narrative is consistent, the conclusions are confirmed by the results. The conclusions should have been spelled out more clearly. A small correction of the English language is required. It is recommended to accept the article after minor changes.
1) Lines 104, 125, 224, 311, 331, 384, 411, 471, 478 and 483. There is an unnecessary dot at the beginning of the line.
Dots are removed.
2) Pages 3, 5 and 8. There is too much free space at the end of the page.
Free spaces are removed.
3) Line 88. It is advisable to remove the hyperlink from the text by placing it in the list of literary sources.
Done (Lines: 83, 419, 472, 488).
4) Line 501. The dot at the end of the sentence is missing.
Dot is added.
5) Line 62. It is advisable to replace the not quite appropriate word "captive". In general, the description of the island archipelago should be more related to the topic of the article.
Desciption of the island archipelago was rewritten (Lines 61-66).
6) Chapter 4.1.2. With the help of a multiscan, measurements can be carried out every hour. Why was such a scheme chosen – once every 3 hours and at the end, after 24 hours? It would be possible to get a more detailed picture, almost without additional effort, simply by adding a protocol.
We totally agree with this reviewer's comment. Unfortunately, no temperature-controlled plate reader was available in our laboratories, so we had to plan our study with manual readings The choice of these time points every 3 hours until 9 hours was chosen because the preliminary results indicated that it was not necessary to carry out measurements every hour. After 9 hours, however, it would be interesting to have a more precise follow-up. However, we have a microplate reader with a manual reading, so it was complicated to perform growth monitoring beyond 9 hours. For the 24-hour reading, it was therefore carried out the next day upon returning to the laboratory. In the event that the breakpoint was greater than 9 hours, we could have launched the microplates in the evening and started monitoring the following morning to obtain the values between T10 and T24, but this was not done.
7) Chapter 4.1.2. 96 holes would allow more concentrations to be used. Why did the authors choose only two concentrations?
Two concentrations have been tested for simplicity of handling and are sufficient to differentiate CP-CRE, Non-CP-CRE and CRE. However, in the discussion section, we propose to add other concentrations or other combinations of antibiotics to distinguish the different carbapenemases within the CP-CRE. Using two concentrations plus the control also allows a larger number of samples to be plated on the microplate (32 including controls in our design).
8) Figures 1 and 2 could be drawn not in the form of roadblocks, but in the form of curved lines, and three lines can be placed together on one graph. Or more than three if the authors took more than two concentrations. It would be more visual.
We propose to keep the form of roadblocks but to move Figure 1b in supplementary data and to delete Figure 2 to make it clearer. This decision was made to also take into account another reviewer’s comment.
9) The conclusions of the entire study should be written more specifically. In chapter 2, the conclusions are formulated too concretely, in chapter 3, basically, the situation as a whole is described. It is advisable to add to chapter 3 a few generalizing suggestions concerning only the work of the authors. Chapter 5 should be rewritten, leaving in it specific conclusions from the study.
We have taken this into account and propose new formulations in the revised MS.
Round 2
Reviewer 1 Report
The authors well responded to the queries and suggestions.
This reviewer has no any comments.